# Influence of Lower Tropospheric Moisture on Local Soil Moisture-Precipitation Feedback over the U.S. Southern Great Plains

Gaoyun Wang[1,2,3], Rong Fu[2], Yizhou Zhuang[2], Paul A. Dirmeyer[4], Joseph A. Santanello[5], Guiling Wang[6], Kun Yang[7], Kaighin McColl[8]

[1]Department of Atmospheric and Oceanic Sciences, School of Physics, Peking University, Beijing, China
[2]Department of Atmospheric and Oceanic Sciences, University of California, Los Angeles, CA, USA
[3]The High School Affiliated to Southern University of Science and Technology, Shenzhen, China
[4]Department of Atmospheric, Oceanic and Earth Sciences, George Mason University, Fairfax, VA, USA
[5]NASA Godard Space Flight Center, Greenbelt, MD, USA
[6]Department of Civil and Environmental Engineering, University of Connecticut, Storrs, CT, USA
[7]Department of Earth System Science, Tsinghua University, Beijing, China
[8]Department of Earth and Planetary Sciences, Harvard University, Cambridge, MA, USA

*Correspondence to*: Yizhou Zhuang (zhuangyz@atmos.ucla.edu)

**Abstract.** Land-atmosphere coupling (LAC) has long been studied focusing on land surface and atmospheric boundary layer processes. However, the influence of humidity in the lower troposphere (LT), especially that above the planetary boundary layer (PBL), on LAC remains largely unexplored. In this study, we use radiosonde observations from the U.S. Southern Great Plains (SGP) site and an entrained parcel buoyancy model to investigate the impact of LT humidity on LAC there during the warm season (May-September). We quantify the effect of LT humidity on convective buoyancy by measuring the difference between the 2-4 km vertically integrated buoyancy with and without the influence of background LT humidity. Our results show that, under dry soil conditions, anomalously high LT humidity is necessary to produce the buoyancy profiles required for afternoon precipitation events (APEs). These APEs under dry soil moisture cannot be explained by commonly used local land-atmosphere coupling indices such as the convective triggering potential/low-level humidity index (CTP/HI$_{Low}$), which do not account for the influence of the LT humidity. On the other hand, consideration of LT humidity is unnecessary to explain APEs under wet soil moisture conditions, suggesting the boundary layer moisture alone could be sufficient to generate the required buoyancy profiles. These findings highlight the need to consider the impact of LT humidity, which is often decoupled from the humidity near the surface and largely controlled by moisture transport, in understanding land-atmospheric feedbacks over dry soil conditions, especially during droughts or dry spells over the SGP.

## 1 Introduction

Land-atmosphere coupling (LAC) plays an important role in determining local and regional climate variability, including surface temperature, humidity, cloud, precipitation, and climate extremes such as drought and floods, especially during the warm season over interior continents (e.g., Fernando et al., 2016; Koster et al., 2004, 2006; Guo et al., 2006; Wang et al., 2007; Roundy and Santanello, 2017; Santanello et al., 2009; Konings et al., 2010; Song et

al., 2016; Roundy et al., 2013). To provide a consistent characterization of land-atmosphere coupling, the International Global Energy and Water Exchanges Project (GEWEX) developed the Local Land-Atmosphere Coupling (LoCo) initiative to coordinate and promote process-level metrics that quantify and characterize LAC (Santanello et al., 2018). The LoCo initiative develops a suite of integrative metrics to quantify the complex relationships and feedback between

the land surface and atmosphere. For example, the mixing diagram approach (Santanello et al., 2009) relates the daytime coevolution of 2-m potential temperature and humidity to the energy and water budgets and growth of the planetary boundary layer (PBL). The convective triggering potential/low-level humidity index (CTP/$HI_{Low}$; Findell & Eltahir, 2003) characterizes the lower tropospheric lapse rate and dewpoint depression of the PBL for convection. The heated condensation framework (HCF, Tawfik et al., 2015a, 2015b) diagnoses the contribution of surface fluxes

to convective initiation based on temperature and humidity profiles. The soil moisture-precipitation (SM-P) feedback is one of the most extensively studied land-atmospheric feedbacks in the literature (e.g., Koster et al., 2004; Ferguson & Wood, 2010; Roundy & Santanello, 2017; Santanello et al., 2018), particularly regarding its effects on the frequency and intensity of convective precipitation (e.g. Taylor, 2015; Tuttle & Salvucci, 2016; Yin et al., 2015).

LoCo investigates the links in the chain coupling soil moisture with the PBL, which connect through surface fluxes,

2-meter temperature and humidity, PBL growth and entrainment, cloud, and precipitation. However, the humidity in the lower troposphere (LT) above the PBL, i.e., ~2-4 km above ground level (AGL), is not explicitly included in previous research. Recent research indicates that specific humidity in the LT ($q_{LT}$) plays a central role in triggering (or the development of) deep convection in the tropics, subtropics, and mid-latitudes during the warm season (Bretherton et al., 2004; Holloway and Neelin, 2009; Zhang and Klein, 2010; Zhuang et al., 2018) and in the

convective initiation driven by land surface heating (Tawfik et al., 2015b, a). The lateral entrainment of $q_{LT}$ dominates buoyancy above the PBL, which is crucial for deep convection development, while entrainment of air at the cloud base has a stronger influence on shallow convection (Holloway and Neelin, 2009; Mapes et al., 2006).

A moist LT can enhance convection by entraining moist air plumes, while low LT humidity can dilute moist plumes originating from the PBL, thereby interfering with the surface influence on convection and precipitation. Thus, $q_{LT}$

determines whether shallow convection can develop into deep convection locally (Schiro et al., 2016; Zhang and Klein, 2010; Zhuang et al., 2017, 2018) and the occurrence and intensity of mesoscale convection (Schiro et al., 2018). $q_{LT}$ is influenced by moisture transport from the PBL (which is largely influenced by land surface), horizontal moisture advection, and subsidence that mixes dry air from aloft. Therefore, it is important to study the relative influences of land surface versus large-scale atmospheric circulation on rainfall and clouds.

The Department of Energy's Atmospheric Radiation Measurement (DOE ARM) project has been pivotal in providing comprehensive datasets for investigating land-atmospheric interactions over the past two decades (e.g., Zhang and Klein, 2010; Santanello et al., 2018). Among the various ARM sites, the Southern Great Plains (SGP) site stands out as the project's inaugural site and one of the most heavily instrumented sites. The SGP region is also widely known as a hotspot of land–atmosphere interactions, as evidenced by numerous past research (e.g., Wakefield et al., 2019;

Santanello et al., 2018; Dirmeyer et al., 2006; Koster et al., 2004; Koster et al., 2006; Guo et al., 2006). This study aims to quantify the impact of LT humidity on the SM-P relationship and local LAC at the SGP site by utilizing an entrained parcel buoyancy model (Zhuang et al. 2018) and the correlation between LT humidity and near-surface

humidity. The dataset and methods are described in section 2. The results are reported in section 3. Discussion and conclusions are provided in section 4.

## 2 Data and method

### 2.1 Dataset

This study focuses on the local warm season (May-September) when thermodynamically driven convection is most prevalent and land-surface feedback plays an important role in determining precipitation (Myoung and Nielsen-Gammon, 2010). Unless stated otherwise, all measurements are taken at the DOE ARM SGP central facility (CF) in north-central Oklahoma (36.60°N, 97.48°W), and the region within a 50-km radius of the CF for 2001-2018. Below are the specific details about the datasets used in this study.

#### 2.1.1 Sounding profiles

Sounding profile data at the SGP CF were obtained through balloon sonde observation. This data is available four times daily at 05:30, 11:30, 17:30, and 23:30 local standard time (LST). We only use the 11:30 LST sounding data as it best represents the precondition of afternoon convection. Because the vertical levels vary with each sounding, data were re-gridded into a uniform vertical resolution of 20 m to facilitate composite analysis. Profiles of the dry-bulb temperature ($T$), dew point temperature ($T_d$), and atmospheric pressure ($p$) were used to calculate the mixing ratio ($r$), specific humidity ($q$), and buoyancy ($b$) using the entrained parcel model (described in section 2.2.1). The data used are available online at https://www.arm.gov/capabilities/instruments/sonde.

#### 2.1.2 Soil moisture

Fractional water index (FWI) is a normalized measurement specifically developed for the Campbell 229-L soil moisture sensor and ranges from 0 for very dry soil to 1 for saturated soil (Schneider et al., 2003). FWI can capture soil wetness independent of soil texture, so it standardizes the observation and allows for intercomparison among different sites with different soil types. Most root biomass in the SGP region and its vicinity is within the top 30 cm of the soil profile (e.g., Raz-Yaseef et al., 2015). Because evapotranspiration, a vital link in the SM-P relationship, is heavily influenced by plant and root zone soil moisture, we used FWI at 25 cm measurement depth provided by the Oklahoma Mesonet Soil Moisture (OKMSOIL) value-added product (VAP) (available at https://www.arm.gov/capabilities/vaps/okmsoil). This data has a 30-min resolution, and we use the average FWI during 06:00-12:00 LST to represent soil moisture condition before afternoon precipitation at daily scale. Wet soils are defined as those with FWI greater than 0.7, which is considered optimal for the plant, and dry soils are defined as FWI smaller than 0.4, which could result in water stress and plant wilting (Illston et al., 2008; Wakefield et al., 2019).

#### 2.1.3 Precipitation

The Arkansas-Red Basin River Forecast Center (ABRFC) precipitation data is based on WSR-88D Nexrad radar precipitation estimates and rain gauge reports with extensive quality control (Fulton et al., 1998). This is an hourly

gridded data product and is available at https://www.arm.gov/capabilities/vaps/abrfc. We used spatially averaged data over the region within a 50 km radius of the SGP CF for this study.

### 2.1.4 PBL height

PBL height data are obtained from the ARM's Planetary Boundary Layer Height (PBLHT) value-added products derived from radiosonde data using the algorithm developed by Liu and Liang (2010). This data is available at
https://www.arm.gov/capabilities/vaps/pblht.

### 2.2 Quantifying contributions of surface and LT humidity to convective buoyancy

Previous research on local land-atmosphere interaction mainly focused on the influence of surface flux and moisture in the PBL on convection initiation and precipitation, such as those related to HCF and mixing diagram metrics. The CTP/$HI_{Low}$ metric considers the effect of lapse rate 100-300 hPa (or about 2-4 km) above ground level (AGL) on
vertically integrated buoyancy in the LT (i.e., CTP) and humidity of the PBL (i.e., $HI_{Low}$), but it does not account for the impact of LT moisture.

Isolating the local influence from other factors in observation presents a significant challenge. Understanding the relationship between near-surface and upper-level information could be crucial to address this. In this study, we first examine the correlation between specific humidity ($q$) profile in the LT and the mixed-layer humidity ($q_m$), defined as
average $q$ in the 0-1 km AGL mixed layer, to assess the potential influence of land surface on LT moisture (Figure 1). We choose mixed layer humidity over humidity directly above surface to represent land surface moisture condition because: 1) radiosonde measurements near the surface are often more susceptible to errors and local disturbances, which could skew the representation of actual surface moisture condition; 2) at noon, 0-1 km mixed layer offers a more representative snapshot of the land surface moisture by capturing the integrated effect of surface evaporation
and convective mixing process; 3) we observe strong correlations, exceeding 0.95 (p<0.05), between the $q$ near the surface and $q_m$. However, this correlation diminishes with increasing height above the PBL. Notably, the LT humidity above 2 km maintains a significant correlation with $q_m$, suggesting a potential influence from the surface. To isolate the effect of land surface on LT humidity, we establish a "land-coupled LT humidity profile $q_{LC}$" for 2-4 km AGL, which is linked to land surface moisture condition. This profile is derived using a linear regression between $q(h,t)$
profile within this layer (2 km $\leq h \leq$ 4 km) and $q_m(t)$. In our regression model, represented by the equation $y = a \times x + b$, $y$ is $q$ at a given height and time $q(h,t)$, and $x$ is $q_m(t)$, with $a(h)$ and $b(h)$ being the linear coefficients at each height level. By solving $a(h)$ and $b(h)$ for each height level in the LT, we can then calculate the "land-coupled LT humidity" as the fitted LT humidity, i.e., $q_{LC}(h,t) = \hat{q}(h,t) = a(h)* q_m(t) + b(h)$.

To quantify the direct influence of LT moisture on convective buoyancy, we adopt an entraining parcel model used in
Zhuang et al. (2018). In this model, the air parcel is lifted with the initial condition of average value within the mixed layer. The ascending air parcel then goes through three processes at each vertical level: dry adiabatic process (parcel ascends without interacting with the environment, entropy conservation), entrainment process (interacts with ambient air, enthalpy conservation), precipitation process (releases condensate, temperature conservation). We apply the deep inflow-A entrainment (DIA) scheme which has been shown to more realistically represent the buoyancy profile

required for deep convection compared to the other assumptions of the lateral entrainment rates such as the constant fractional entrainment rate scheme (e.g., Holloway & Neelin, 2009; Schiro et al., 2016; Siebesma et al., 2007). The DIA scheme uses an entrainment rate inversely proportional to the altitude ($\alpha z^{-1}$) for simplicity. All condensates formed from the previous two processes are set to fall out in the precipitation process (pseudo-adiabatic process). And finally, buoyancy is calculated by $b = g \frac{T_{pv} - T_{ev}}{T_{ev}}$, where $T_{pv}$ and $T_{ev}$ are the virtual temperature of the parcel and the

environment, respectively. Details of this model are provided in Zhuang et al. (2018).

To quantify the influence of LT humidity on the lateral entrainment of the convection and, consequently, the buoyancy of the convective air parcel, we consider two $q$ profiles for the lateral entrainment process: 1) the observed humidity profile ($q_R$) and 2) the land-coupled humidity profile ($q_{LC}$). The $q_{LC}$ below 2 km AGL equals to the observed $q$, and $q_{LC}$ between 2 km and 4 km AGL is a coupled LT humidity profile constructed from the regression between $q$ and

averaged $q$ in the mixed layer (as described in 2.2). Since the effect of entrainment accumulates continuously after the parcel is lifted, we calculate the buoyancy profile with $q_R$ as humidity profile ($b_R$) and that with $q_{LC}$ as humidity profile ($b_{LC}$), and then use the vertical integral of their difference in the LT (2-4 km AGL), $B_{LT} = \int_{2\,km\,AGL}^{4km\,AGL}(b_R - b_{LC})dz$, to quantify the additional effect of LT moisture variation on the parcel buoyancy that is not coupled with the PBL. We also calculate the integral of buoyancy based on $q_{LC}$, i.e., $B_{LC} = \int_{2km\,AGL}^{4km\,AGL} b_{LC}dz$, to assess

the land-coupled effect on convection. To make the results comparable, we apply a normal percentile transform (Wilks, 2011) to obtain standardized scores of $B_{LT}$, which we use for further analysis.

**2.3 Identify dry/wet soil regime and coupled afternoon precipitation events**

The CTP/HI$_{Low}$ framework developed by Findell & Eltahir (2003) is commonly used to identify atmospheric preference of LAC state. CTP is calculated by integrating the difference between moist adiabat temperature and the

ambient temperature profile from 100 to 300 hPa AGL. It is a measure of the energy available for convection and the 100–300 hPa AGL is a critical level for the development of the daytime boundary layer. HI$_{Low}$, on the other hand, indicates the pre-existing moisture of the very lower atmosphere, and is defined as HI$_{Low}$ $= (T - T_d)_{150hPa\,AGL} + (T - T_d)_{50hPa\,AGL}$. In this framework, wet soil advantage regime occurs when the atmospheric state is closer to the wet adiabatic rate, resulting in a low CTP and large latent heat flux (small HI$_{Low}$). Conversely, dry soil advantage

regime occurs when the temperature profile is close to the dry adiabatic lapse rate with weak thermal stability (high CTP), and the soil provides less water vapor but more heat. This condition favors convection lifted by the boundary layer growth due to high sensible heat fluxes at the surface (Ek and Holtslag, 2004; Huang and Margulis, 2011; Gentine et al., 2013).

We adopt a modified CTP/HI$_{Low}$ framework proposed by Wakefield et al. (2019) using the standardized score of

CTP/HI$_{Low}$. We first calculate CTP and HI$_{Low}$ using sounding data at 11:30 LST, and average FWI during 06:00-12:00 LST. Then dry-coupling cases are defined as days with anomalously high CTP (higher than climatological CTP for our analysis period) over dry soil (FWI < 0.4), which is similar to the dry soil advantage regime in Findell & Eltahir (2003); wet-coupling cases, on the other hand, are characterized by anomalously low HI$_{Low}$ over wet soil (FWI > 0.7), which corresponds to a moisture-abundant, energy-limited regime where the atmospheric profile is likely near

175 moist adiabatic (Findell and Eltahir, 2003). The wet soil condition is expected to promote precipitation recycling through the addition of moist static energy via evapotranspiration and provides a continuous supply of low-level moisture.

Since LAC would mostly affect the thermodynamically driven afternoon convection, we focus on the morning (0600-1300 LST), afternoon (1400-2000 LST), and evening (2100-2400 LST) precipitation events in our analysis. Afternoon
180 precipitation events (APEs) are identified as daily samples that meet the following two criteria: 1) daily precipitation peaks during the afternoon hours defined above; and 2) the afternoon precipitation is at least twice as large as the morning precipitation, and also greater than the evening precipitation (filter out organized precipitation). The cases not categorized as APEs are referred to as non-APEs afterward. We obtain a total of 368 APEs from the 2172 sounding. We further select APEs associated with either dry-coupling or wet-coupling condition, resulting in 94 dry-coupling
185 APEs and 79 wet-coupling APEs. These account for 24.2% of the total 388 dry-coupling cases and 20.3 % of the total 389 wet-coupling cases, respectively. The comparable number of APEs for both dry- and wet-coupling conditions aligns with the finding of Findell and Eltahir (2003b) that the SGP is in the transitional region where negative and positive feedback days occurred with similar frequency. In addition, our analysis also shows that (Figure S1), within all 368 APEs, 16 instances exhibit a $HI_{Low}$ lower than 5°C — a threshold established in Findell and Eltahir (2003a,
190 2003b). Among these, 8 are wet-coupling APEs and have significantly higher FWI compared to other groups. This suggests that the low $HI_{Low}$ values observed before noon in these cases are likely influenced by soil moisture evaporation, rather than being purely controlled by atmospheric factors. Furthermore, one of these cases is categorized as dry-coupling APE, and seven as "other APEs", which are APEs not categorized as either dry-coupling or wet-coupling APEs. These cases likely represent "atmospherically controlled days", as per the CTP- $HI_{Low}$ framework, and
195 only account for a small fraction (~2.2%) of all APEs we identified.

**3 Results**

**3.1 Thermodynamic pre-conditions for APEs under the dry and wet-coupling regimes**

To identify favorable atmospheric conditions for coupling APEs, we evaluate the differences in temperature ($T$), specific humidity ($q$), and relative humidity (RH) at 11:30 LST between averaged local coupling APEs and non-APE
200 cases in the warm seasons (May to September), as shown in Figure 2. Regardless of soil moisture conditions and coupling regimes, APEs are always associated with a wetter lower troposphere (0-4 km) than non-APEs. For dry-coupling regimes, the increases of $q$ and RH in the PBL and the LT associated with APEs are also stronger than those with the non-APEs, especially between 0.5 km and 3.5 km AGL, and the contrast between APEs and non-APEs in dry-coupling regime is larger than that for wet-coupling regimes. These humidity differences are expected, as more
205 humid pre-conditions favor the occurrence of APEs. Notice that the greatest contrast between RH of APEs and that of non-APEs occurs between 1 km and 3.5 km AGL (above the PBL), which is the combined result of high $q$ and decreasing $T$ over this layer, highlighting the possible strong influence of LT humidity on deep convection.

In contrast, the sign of the temperature difference between APEs and non-APEs below 1.7 km AGL varies between the dry-coupling and wet-coupling regimes. For the dry-coupling regimes, the average temperature of APEs is lower

than that of the non-APEs. This is presumably due to the stronger surface sensible flux and less stable atmosphere (a steep lapse rate or faster decrease of temperature with height) associated with APEs than the non-APEs under the dry-coupling regime. For the wet-coupling regime, the average temperature of the APEs is warmer than that of the non-APEs below 1.7 km AGL. This is consistent with a weaker lapse rate associated with a more humid environment, presumably due to vertical mixing of shallow convection, in the APEs than in the non-APEs cases. Above 1.7 km

AGL, temperature of the APEs cases is lower than that of the non-APEs for both dry and wet-coupling regimes, as expected from less stable thermodynamic conditions in the APEs than in the non-APEs.

To investigate the difference in atmospheric conditions that favor APEs under the dry- versus wet-coupling regime, we compare the composite differential profiles of RH, $q$, and $T$ between dry- and wet-coupling APEs, as shown in Figure 3. In general, $T$ is higher for APEs under dry-coupling than under wet-coupling regime (Figure 3a), especially

in the PBL (below 2 km). This can be attributed to stronger sensible heat flux and temperature mixing over a dry surface. Notably, there is a significant difference in LT specific humidity between dry- and wet-coupling regimes (Figure 3b), with $q$ associated with APEs being slightly lower below 1 km AGL under dry-coupling than under wet-coupling regime, but higher above 1 km, especially between 2 km and 3 km AGL. This suggests that APEs require entrainment of higher LT moisture under dry-coupling than under wet-coupling regimes. Both Figures 2b and 3b

suggest that higher LT specific humidity is needed for APEs under the dry-coupling than under the wet-coupling regimes. Moreover, RH associated with dry-coupling regimes is less than that of wet-coupling regimes in the PBL (Figure 3c), as expected from a drier PBL over a dry surface. However, such an RH difference becomes smaller and eventually disappears in the LT (2-4 km). This is because the lower RH in the dry-coupling regimes is mainly due to warmer $T$ below 2 km AGL (Figure 3b), whereas above 2 km AGL, the higher $q$ and slightly warmer $T$ in the dry-

coupling regimes balance each other out and lead to a similar RH as in the wet-coupling regimes.

Recent studies on the SM-P relationship have highlighted the greater impact of soil-moisture anomalies on boundary-layer stability and precipitation formation than on the ambient moisture (e.g., Seneviratne et al., 2010; Santanello et al., 2018). To investigate how humidity affects the preconditioning of the convective environment and how it impacts instability in dry- and wet-coupling regimes, we use an entraining parcel model to calculate buoyancy profiles for $b_R$

and $b_{LC}$, respectively, with parcels originating in the mixed layer. We then compute differences in the integral buoyancy between $b_R$ and $b_{LC}$ profiles for the 2-4 km AGL range to explore the influence of LT humidity-related convective thermodynamic instability.

To evaluate the atmospheric thermodynamic structure associated with the $B_{LT}$, we evaluate the composite average sounding profiles based on three tertile of $B_{LT}$ in the warm season as shown in Figure 4. The $B_{LT}$ values for the three

terciles range from -55.8 to -8.8 J/kg, -8.8 to 10.20J/kg, and 10.0 to 72.8 J/kg, respectively. It is noteworthy that the temperature and dew point are similar among these three terciles of $B_{LT}$ near the surface (below 900hPa) but clearly different from above 900 hPa up to at least 400 hPa AGL, which indicates the importance of LT humidity.

For the lower tercile (0% - 33%) of $B_{LT}$, dew point values are substantially lower than those for the middle and upper tercile of the $B_{LT}$. The sharp decrease of dew point values with height, near-constant temperature, and large gap

between the temperature and dew point profiles at 700-900 hPa suggest strong dry shallow convection. For the middle tercile, dewpoint and temperature decrease gradually with a height between 900 hPa and 700 hPa, and the gap between

the dew point and temperature profiles is smaller than those for the lower tercile of the $B_{LT}$. These features suggest a mixture of dry and moist shallow convection. For the upper tercile of $B_{LT}$ (67% – 100%), dew point values are nearly constant from the surface to 700 hPa, and the humidity of free troposphere, as indicated by the gap between temperature and dew point profiles, is substantially wetter (implying higher RH) than for the middle and lower tercile of the $B_{LT}$. These features suggest that moist shallow convection dominates LT. Higher dew point values between 700 hPa and 500 hPa also suggest a more humid middle troposphere associated with the upper $B_{LT}$ tercile than with the middle and lower $B_{LT}$ tercile. Thus, Figure 4 suggests that $B_{LT}$ variations are strongly influenced by the humidity in the lower and middle troposphere.

### 3.2 The influence of LT humidity on afternoon precipitation under different LAC regimes

To investigate the effect of the LT humidity versus land surface air humidity on APEs, we evaluate the probability distribution of APEs as a function of $B_{LT}$ and $B_{LC}$ for dry- and wet-coupling APEs, respectively, in Figure 5. $B_{LC}$ is usually more negative over dry soil (Figure 5b) than over wet soil (Figure 5a), and therefore usually more negative over dry PBL than over wet PBL. Figure 5b shows that dry-coupling APEs occur more frequently with negative $B_{LC}$ (69%) and positive $B_{LT}$ (77%), indicating that the influence of a more humid LT can override the influence of surface air aridity. For land surface, favoring local precipitation ($B_{LC}$ >0), dry-coupling APEs occur more frequently with positive $B_{LT}$ (26%) than with negative $B_{LT}$ (5%). For the wet-coupling conditions, 57% of APEs occur with positive $B_{LT}$, while 31% occur with humid PBL pre-condition (Figure 5a). Overall, for dry-coupling cases, more APEs are associated with humid LT (positive $B_{LT}$) than with humid surface air (positive $B_{LC}$), but this trend is not evident for wet-coupling cases.

To evaluate the influence of LT humidity on LAC under the CTP/HI$_{Low}$ framework, we compare the joint distribution of all APEs, wet-coupling APEs, and dry-coupling APEs, respectively, as a function of $B_{LT}$ scores and HI$_{Low}$ scores (Figure 6) using a normal percentile transform (Wilks 2011). Larger than normal $B_{LT}$ (humid LT) values were associated with 61% of all APEs, regardless of the PBL humidity. Smaller than normal HI$_{Low}$ (humid PBL) occurred in 63% of all APEs, regardless of the LT humidity. About 40% of all APEs occur under both humid PBL and humid LT. Thus, the probabilities for APEs to occur under either humid PBL or humid LT are similar, with a preference for APEs to occur under both humid PBL and humid LT conditions. For dry-coupling condition, 76% of the APEs occur under humid LT versus 59% under humid PBL. Therefore, APEs appear to prefer humid LT more than humid PBL under dry-coupling conditions, but this preference is not found in wet-coupling conditions. This result is consistent with the findings in Figure 5.

To further investigate how LT humidity or $B_{LT}$ can affect the probability of APEs under the dry-coupling and wet-coupling regimes, respectively, we present three statistical measures of APEs as a function of $B_{LT}$ in Figure 7. Figure 7a shows that, for the dry-coupling cases, the fractional occurrence of APEs (defined as the proportion of APEs relative to all dry-coupling cases) in each $B_{LT}$ bin increases with $B_{LT}$ up to its 70[th] percentile, with a significant correlation ($R$ = 0.65, $p < 0.05$). For the wet-coupling cases, the fractional occurrence of the APEs peaks when $B_{LT}$ is between the 30[th] and 70[th] percentile. Thus, APEs appear to prefer higher LT humidity under dry coupling than under wet coupling.

Next, we explore how $B_{LT}$ affects the partition of dry-coupling versus wet-coupling APEs. Figure 7b shows that the proportion of the dry-coupling APEs relative to all APEs increases with $B_{LT}$, with a strong correlation ($R = 0.89$, $p$ <0.05). The proportion ranges from 0.04 at the bottom 10% to 0.47 at the top 10% of $B_{LT}$. However, the proportion of

wet-coupling APEs per $B_{LT}$ bin peaks at lower to medium $B_{LT}$ percentiles (30%-50%) and decreases almost monotonically with increasing $B_{LT}$ from 50% to 100%.

We also investigate how $B_{LT}$ affects rain rates associated with dry-coupling and wet-coupling APEs, respectively. Figure 7c shows a clear increase in rain rate with $B_{LT}$ for the dry-coupling APEs, except for the 90-100[th] percentile of $B_{LT}$, where there are few APEs samples. In contrast, we find no clear dependence of rain rate on $B_{LT}$ for the wet-

coupling APEs. Thus, our findings suggest that a high $B_{LT}$ tends to increase the frequency and intensity of the dry-coupling APEs, as well as the relative frequency of dry-coupling APEs compared to wet-coupling APEs.

In addition, we evaluate the variations of deep- (cloud top height (CTH) > 8 km), shallow- (CTH <3km), and convective congestus (CTH between 3 km and 8 km) associated with APEs based on hourly precipitation and cloud fraction following Zhuang et al. (2017). In general, APEs associated with all three convective types increase with $B_{LT}$

under dry-coupling conditions (Figure S2). Under wet-coupling condition, APEs associated with deep convection does not exhibit a clear dependence on $B_{LT}$. However, APEs associated with shallow convection decreases with increasing $B_{LT}$, while those associated with congestus increase with increasing $B_{LT}$. These results imply that the increase in $B_{LT}$ can lead to a deepening of shallow convection into congestus due to reduced buoyancy dilution caused by entraining wetter LT air for wet-coupling convection.

**4 Conclusions**

Land-atmosphere interactions occur when local land surface and subsurface conditions influence the moisture and energy budgets of the overlying atmosphere. The relative impacts of soil moisture on convective precipitation can vary depending on the atmospheric conditions. In this study, we compared the difference in RH, $q$, and $T$ profiles between APEs and non-APEs under both dry- and wet-coupling conditions.

Our initial analysis revealed that APEs had an overall wetter PBL and LT (0-4km AGL) than non-APEs, especially under dry-coupling regimes. The RH difference between APEs and non-APEs in the LT was driven by differences in both $q$ and $T$, with dry-coupling APEs exhibiting lower humidity in the PBL than wet-coupling APEs. However, as the altitude increases, the difference in RH between dry- and wet-coupling APEs decreases due to the increasing difference in $q$ and decreasing difference in $T$. Above 4 km AGL, the difference in $q$ becomes zero. Therefore, we

could infer the importance of LT humidity in the SM-P relationship, and the APEs under dry-coupling conditions necessitate more LT humidity than that under wet-coupling conditions.

To further investigate the influence of LT humidity on the SM-P relationship, we employ an entraining parcel model and a new metric $B_{LT}$, which measures the 2-4 km vertical integral of the difference between buoyancy calculated from the observed humidity profile and that correlated to (regressed against) the average specific humidity in the PBL.

Statistical analysis reveals that the wetter LT and normal PBL were associated with larger $B_{LT}$ values, whereas drier LT was linked to smaller $B_{LT}$ values. Moreover, there is a higher likelihood of APEs occurring with positive $B_{LT}$ percentile under dry-coupling conditions, while this relationship is not apparent for the probability distribution of $B_{LT}$

percentile for wet-coupling APEs. Additionally, as the $B_{LT}$ percentile increases, the frequency of dry-coupling APEs also increases, whereas the opposite tendency was observed for wet-coupling APEs. In the meantime, the ratio of dry-coupling APEs to all APEs increases with the $B_{LT}$ percentile, while this tendency is the opposite for the ratio of wet-coupling APEs to all APEs. Regarding precipitation, the average rain rate tends to rise with increasing $B_{LT}$ percentile under dry-coupling conditions, but this trend is not significant for wet-coupling APEs. Overall, our results indicate that the impact of LT humidity differs between dry- and wet-coupling APEs, with dry-coupling APEs being more influenced by LT humidity compared to wet-coupling APEs.

The Great Plain Low-Level Jet (GPLLJ) is widely acknowledged as a primary mechanism responsible for the regional-scale water vapor transport from the Gulf of Mexico during May-September. The GPLLJ creates a thermodynamic environment that facilitates convection and precipitation, making it a key factor in initiating and sustaining mesoscale weather phenomena (e.g., Higgins et al., 1997; Hodges & Pu, 2019; Mo et al., 1997; Pu et al., 2016; Pu & Dickinson, 2014; Weaver & Nigam, 2008). It is well established that the GPLLJ can enhance the occurrence of nocturnal convective precipitation in the SGP (Pu and Dickinson, 2014). However, our findings imply that the moisture carried by the GPLLJ could also play an important role in generating local diurnal afternoon precipitation when it reaches the region during the daytime, particularly over dry soil conditions. This result is consistent with Ford et al. (2015) thatsoil moisture feedback to precipitation could potentially manifest itself over wetter- and drier-than-normal soils, depending on the overall synoptic and dynamic conditions, and precipitation favors dry soil when the low-level jet is present. Therefore, these results collectively suggest that the GPLLJ plays a significant role in alleviating drought conditions in the SGP by influencing both diurnal and nocturnal precipitation.

Our study not only presents new insights into the role of LT humidity on the SM-P relationship but also serves as a quantitative elucidation of the negative feedback behavior discussed in Findell and Eltahir (2003b) using the CTP-HIlow framework. Specifically, they highlighted the topographic and dynamical circumstances that commonly result in a moist air layer originating from the elevated Mexican plateau, typically with its base around 850 hPa. This moist layer, while not captured by the $HI_{Low}$ metric, can be detected by our approach. However, there are still limitations in our work. One key concern is the potential uncertainties introduced by constructing the land-coupled LT humidity profile via linear regression. Such uncertainties arise mainly from the linear model's inherent assumptions, including the constancy of relationships under varying conditions and the potential oversight of non-linearity. A more thorough investigation into the model's residuals and additional sensitivity analyses could provide deeper insights into these uncertainties. Furthermore, our categorization of APEs may not be always associated with convective precipitation, given that it relies solely on the region-average precipitation data. Improving the classification of APEs, possibly by integrating convection classification results from radar observations, could lead to more precise interpretations. This study only focuses on a single location, i.e., SGP, thus expanding research to include a variety of climatic zones would be crucial in assessing the broader applicability of our methods and conclusions. Our future work will also involve investigating the primary source of LT humidity and employing both $B_{LT}$ and CTP/$HI_{Low}$ as atmospheric indicators to identify global regions with diverse LT humidity-SM-P relationships, thereby advancing our understanding of LAC on a broader scale.

**Code Availability**

Codes developed for generating the results of this study can be provided by the corresponding author upon reasonable request.

**Data Availability**

All data used in this study are openly accessible via https://www.arm.gov.

**Author Contribution**

RF, GW, and YZ designed the research and developed the methodology. GW performed formal analysis and wrote the initial draft. All authors reviewed and edited the draft.

**Competing Interests**

The contact author has declared that none of the authors has any competing interests.

**Acknowledgment**

G.W. was funded by the China Scholarship Council (CSC; 201806010052). Y.Z. and R.F. were supported by the National Oceanic and Atmospheric Administration-Climate Program Office (NOAA-CPO) Modelling, Analysis, Predictions, and Projections (MAPP) Program (NA20OAR4310426), and the National Science Foundation (NSF) Physical and Dynamic Meteorology (PDM) Program (AGS-2214697). P.A.D was supported by the National Aeronautics and Space Administration (NASA) (80NSSC21K1801). J.S. was supported by funding from NASA HQ
and PM Jared Entin.

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

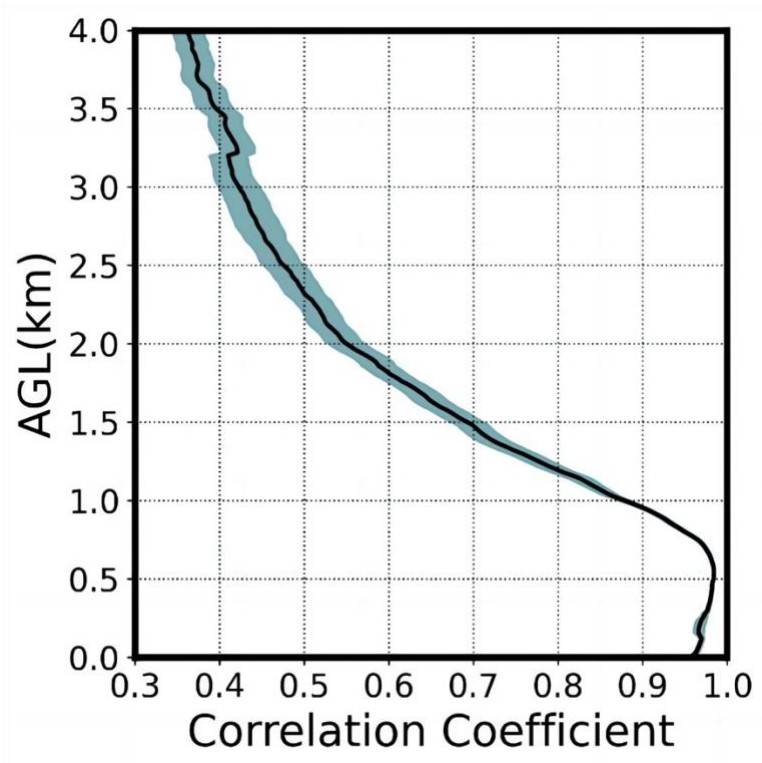

**Figure 1: Correlation coefficient profiles between specific humidity (*q*) at each vertical level from 0 to 4 km AGL and mean**

***q* in the mixed layer (0-1km AGL). The correlation coefficients are calculated for the warm season of each year. The black line indicates the average value of 18 years, and the green shade shows the standard error. All correlation coefficients at 0-4 km for all 18 years are significant at 0.05 level.**

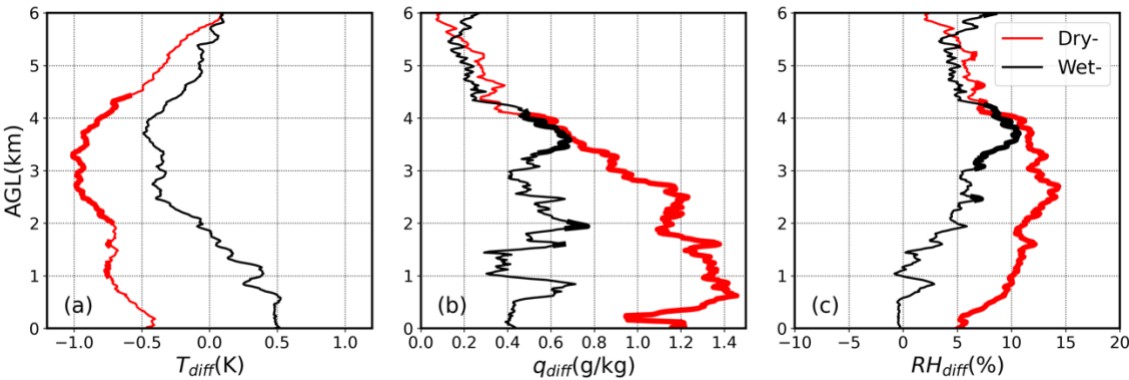

**Figure 2:** Composite difference of a) temperature ($T_{diff}$), b) specific humidity ($q_{diff}$), and c) relative humidity ($RH_{diff}$) profiles between APEs and non-APEs for dry- (red lines) and wet- (black lines) coupling cases. The thicker portions of the lines indicate where the differences are statistically significant at 0.05 level.

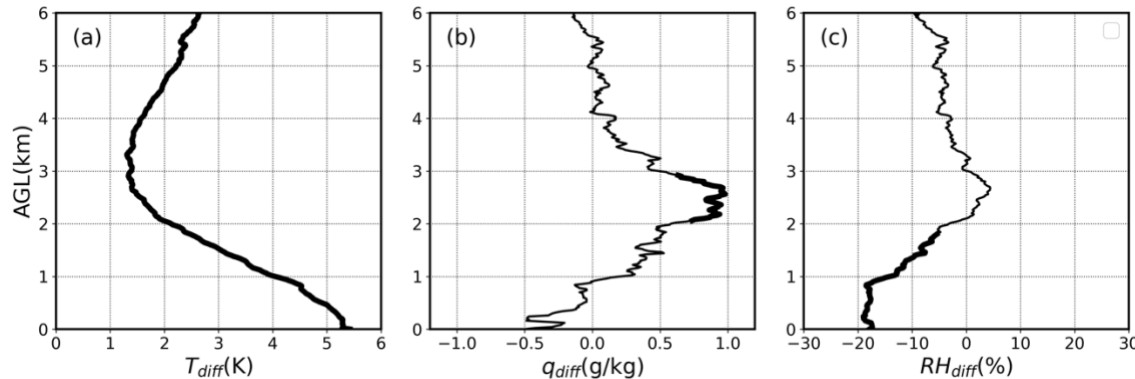

**Figure 3:** Composite difference of a) temperature ($T_{diff}$), b) specific humidity($q_{diff}$), and c) relative humidity ($RH_{diff}$) between dry- and wet-coupling APEs (Dry minus Wet). The thicker portions of the lines indicate where the differences are statistically significant at 0.05 level.

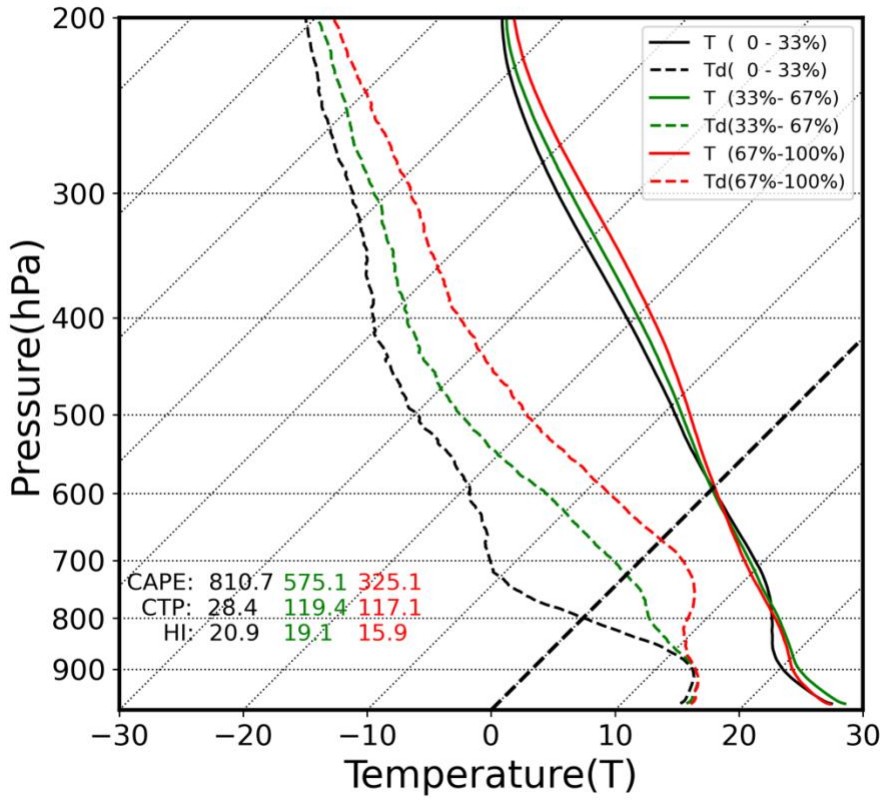


**Figure 4:** Composite temperature ($T$; solid line) and dewpoint temperature ($T_d$; dash line) profiles at the ARM SGP site for all days during May–September from 2001 to 2018, above the 950hPa level, based on $B_{LT}$ tercile: 0-33% (lower $B_{LT}$, black); 33%-67% (medium $B_{LT}$, green); 67%-100% bins (higher $B_{LT}$, red).

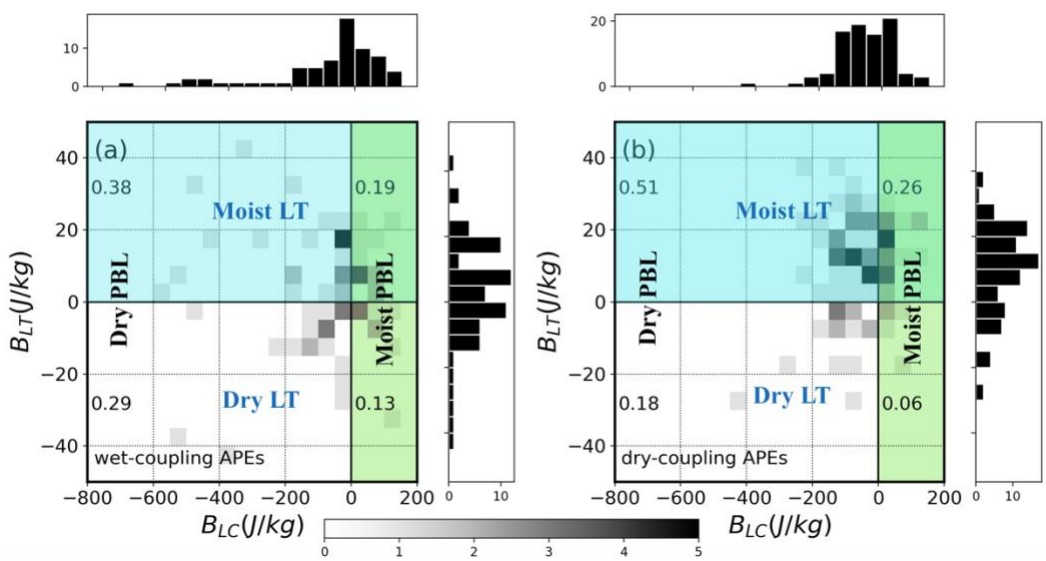


**Figure 5:** Joint frequency distributions of a) wet-coupling APEs and b) dry-coupling APEs as a function of $B_{LT}$ (representing contribution of LT humidity to convective buoyancy) and $B_{LC}$ (representing contribution of surface humidity) with white shades representing no APEs and darkest shades representing more than 5 APEs occurs of each $B_{LC}$ - $B_{LT}$ bin. The number indicates the fraction in each quadrant.


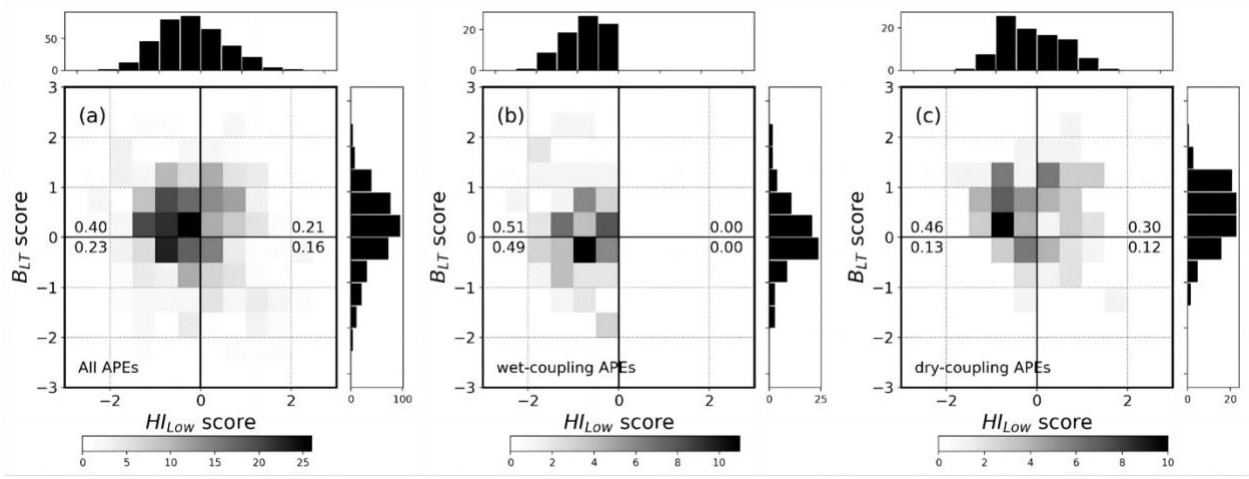

**Figure 6:** Joint distribution of (a) all APEs, (b) wet-coupling APEs, and (c) dry-coupling APEs as functions of $B_{LT}$ score and $HI_{Low}$ score. The number indicates the fraction in each quadrant.


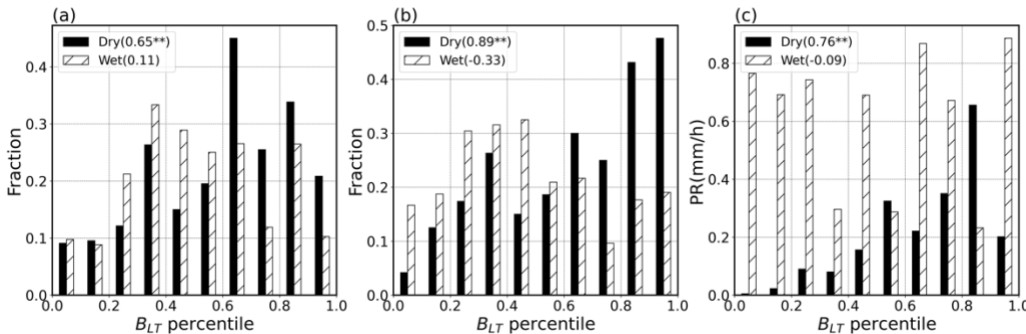

Figure 7: (a) The fraction of wet- (dry-) coupling APEs over wet- (dry-) coupling cases ($\frac{N_{wet-coupling-APEs}}{N_{wet-coupling-cases}}$ or $\frac{N_{dry-coupling-APEs}}{N_{dry-coupling-cases}}$) in each $B_{LT}$ bin for the dry- and wet-coupling cases, respectively. (b) Same as (a), but for the percentage of wet- or dry-coupling APEs relative to all APEs ($\frac{N_{wet-coupling-APEs}}{N_{APEs}}$ and $\frac{N_{dry-coupling-APEs}}{N_{APEs}}$). (c) Same as (a), but for the mean afternoon precipitation rate (PR). The correlation coefficients between $B_{LT}$ percentiles and the fraction/PR are listed for the dry- and wet-coupling cases, respectively; correlation coefficients between $B_{LT}$ percentiles and y-axis value significant at 0.05 level are marked with two asterisks.
