# Peer review of "Influence of Lower Tropospheric Moisture on Local Soil Moisture-Precipitation Feedback over the U.S. Southern Great Plains"

_EGUsphere, 2023_

## Referee Comment (RC2)

Review of "Influence of Lower Tropospheric Moisture on Local Soil Moisture-Precipitation Feedback over the U.S. Southern Great Plains" by Wang et al.

**General comments:**
This paper examines the role of lower troposphere (LT) moisture in land-atmosphere coupling (LAC) using radiosonde data from the US Southern Great Plains (SGP) site. The analysis focuses on afternoon precipitation events (APEs) in the warm season (May– September). It is found that LT moisture has a greater impact on dry-coupling APEs than on wet-coupling APEs. A higher $B_{LT}$, which is a vertically integrated LT buoyancy uncoupled with the PBL humidity, tends to increase both the frequency and intensity of dry-coupling APEs. The paper is overall well-structured and easy to follow, and the findings clarify the importance of warm-season LT moisture in dry-coupling conditions in the SGP. I have a few minor suggestions for the authors to consider.

**Specific comments:**
1. While the Introduction Section provides a detailed review of previous studies of LAC and LT moisture, little is mentioned about the SGP and why this region is selected to study the impact of LT moisture. It is also not clear why ARM SGP radiosonde data were used as the primary data in this work. A brief background information about LAC and LT moisture in the SGP and a clarification of the novelty of the approach would be helpful and informative in the Introduction Section.

2. The concept of the lower troposphere (LT) was brought up very early in the paper, but it is not clearly defined until section 2.2 (line 135). It would be better to clarify the definition of LT in the earlier part of the paper.

3. It would be nice to have a section to discuss the uncertainties associated with the data and/or methodology. For instance, the coupled LT humidity profile is reconstructed by linear regressions. What's the uncertainty of the approach? Are all the APEs convective precipitation? In addition, it also would be interesting to briefly discuss to what extent the approach used here and findings in the SGP can be generalized to other regions.

4. In section 2.1, it would be nice to provide more details about the SGP CF site, such as location (lat, long), data coverage, and why the site was selected.

5. In sections 2.1.1-2.1.3, please add information about the temporal resolution and coverage of the datasets and variables used in the paper. It also would be informative to discuss the error ranges of the data, if possible.

6. Section 2.1.4, can you please provide the equation used to calculate PBL height?

7. Line 116, how do you determine the height of the mixed layer?

8. Section 2.4, are both dry- and wet-coupling defined on the daily time scale?

9. I suggest moving Figure S1 to the main text, as it provides useful information and there is sufficient room for one more figure in the main text.

10. Fig. 1, please consider marking the coefficients that are significant at the 95% confidence level. The caption mentioned "…value of 18 years", but 2001-2019 (line 74) is 19 years.

11. Figs. 2-3, consider marking profiles where the composite differences are significant.

12. Fig. 4, why is the data coverage 2001-2018 instead of 2001-2019?

**Technical corrections:**
1. In lines 216 and 219, there is no "Figure 4a" nor "Figure 4b".

2. In Figure 5, the last label of the x-axis ("200") is too close to the first label of the vertical axis of the bar plot ("0"), so it looks like "2000".

---

## Author Comment (AC1)

**Reviewer 1 (Kirsten Findell):**

**General Comments**

This manuscript nicely quantifies the influence of moisture in the portion of the lower troposphere that is just above the boundary layer on the development of convection over dry or wet soils. The authors use radiosonde observations from the US Southern Great Plains field site along with an entrained parcel buoyancy model. Their results demonstrate an important role for lower tropospheric humidity in convective triggering in dry-coupling convective events; they also show that this is far less important in wet-coupling events.

I have two main points I think should be addressed before this paper is ready for publication. First, I don't think these results should be presented as *contrary* to the CTP-HI$_{low}$ framework, but as a quantification and elucidation of the negative feedback behavior that is discussed at length in Part II of the CTP-HI$_{low}$ framework papers (Findell and Eltahir, 2003b, citation provided below). Second, the process quantification in this paper centers around your reconstructed land-coupled humidity profiles. I think you need to provide much more information (along with figures) about these profiles. I feel like they are at the heart of your story, and yet very little information is provided about them. Are they systematically different from the observed humidity profiles? (For example, are they systematically more humid?) Can you show us a plot of the data that the regression is based on? This seems central to your results, so I feel it is important to be clear and transparent about what information is actually captured by the term that leads to attribution of land-driven coupling. Some of this information is revealed through Figure 5, but I think you need to be explicit about bLC as soon as the concept is introduced.

Re: Thank you very much for these helpful comments and suggestions.

First, we agree that our results are not contrary to the CTP-HI$_{low}$ framework and they are further quantification of the results from the CTP-HI$_{low}$ framework. We have added a few sentences in the last paragraph of the manuscript to discuss this. **Note that all line numbers listed below are referred to the track-change version of the revised manuscript.**

Lines 350-354: "Our study not only presents new insights into the role of LT humidity in LAC and convective precipitation but also serves as a quantitative elucidation of the negative feedback behavior discussed in Findell and Eltahir (2003b) using the CTP-HIlow framework. Specifically, they highlighted the topographic and dynamical circumstances that commonly result in a moist air layer originating from the elevated Mexican plateau, typically with its base around 850 hPa. This moist layer, while not captured by the HILow metric, can be detected by our approach. "

Second, about the reconstructed land-coupled humidity profile, this confusion is likely related to our oversimplified description of the model. We revised the method section accordingly:

Lines 126-146: "Isolating the local influence from other factors in observation presents a significant challenge. Understanding the relationship between near-surface and upper-level information could be crucial to address this.  In this study, we first examine the correlation between specific humidity ($q$) profile in the LT and the mixed-layer humidity ($q_m$), defined as average $q$ in the 0-1 km AGL mixed layer, to assess the potential influence of land surface on LT moisture (Figure 1). We choose mixed layer humidity over

humidity directly above surface to represent land surface moisture condition because: 1) radiosonde measurements near the surface are often more susceptible to errors and local disturbances, which could skew the representation of actual surface moisture condition; 2) at noon, 0-1 km mixed layer offers a more representative snapshot of the land surface moisture by capturing the integrated effect of surface evaporation and convective mixing process; 3) we observe strong correlations, exceeding 0.95 ($p<0.05$), between the $q$ near the surface and $q_m$. However, this correlation diminishes with increasing height above the PBL. Notably, the LT humidity above 2km maintains a significant correlation with $q_m$, suggesting a potential influence from the surface. To isolate the effect of land surface on LT humidity, we establish a "land-coupled LT humidity profile $q_{LC}$" for 2-4 km AGL, which is linked to land surface moisture condition. This profile is derived using a linear regression between $q(h,t)$ profile within this layer (2 km < h < 4 km) and $q_m(t)$. In our regression model, represented by the equation $y = a \times x + b$, $y$ is $q$ at a given height and time $q(h,t)$, and $x$ is $q_m(t)$, with $a(h)$ and $b(h)$ being the linear coefficients at each height level. By solving $a(h)$ and $b(h)$ for each height level in the LT, we can then calculate the "land-coupled LT humidity" as the fitted LT humidity, i.e., $q_{LC}(h,t) = \hat{q}(h,t) = a(h)* q_m(t) + b(h)$."

As described above, the land-coupled $q$ ($q_{LC}$) is the fitted $q$ ($\hat{q}$) in the regression, so $q$ and $\hat{q}$ have the same mean and therefore are not systematically different.

I offer a few additional minor suggestions below. I feel that this paper will be a nice contribution to the literature once these issues are addressed. I believe that all the suggestions qualify as minor revisions. I look forward to seeing this work in print!

Respectfully submitted,
Kirsten Findell

**Specific comments**

- Abstract: I think the abstract would be improved by clarification of what you mean by lower troposphere since HIlow does capture some of the lower troposphere. On line 51 you say "lower troposphere (LT) above the PBL": this distinction should be in the abstract, too.

  We added this in Lines 17-18:

  "However, the influence of humidity in the lower troposphere (LT), especially that above the planetary boundary layer (PBL), on LAC remains largely unexplored."

- To further the point mentioned above about consistency between your results and the CTP-HIlow framework, the discussion of physical mechanisms leading to the negative feedback region shown on the map in Figure 2 of Findell and Eltahir (2003b) focuses on the topographic and dynamical circumstances that commonly lead to the presence of a layer of moist air coming off of the Mexican plateau with its base at about 850 mb. As you note, since HIlow is calculated from the humidity deficit 50 and 150 mb above the ground surface, it might not capture this layer of moist air. Nevertheless, I think the results you find are quite consistent with the process understanding that was enabled by Parts I and II of the CTP-HIlow framework papers. Your sentence about lateral entrainment of moisture on lines 55-57 seems to echo the discussion of processes leading to the negative feedback regime provided in Findell and Eltahir (2003b).

Thank you for furthering the point in the general comment. We have added a few discussion outlined in our reply to your general comment above.

- The SGP site is not in the negative feedback region shown in Findell and Eltahir (2003b), but in the transitional region just to the east. In this region, negative and positive feedback days were shown to occur with approximately the same frequency. This is also consistent with your results.

  Thanks, we added Lines 198-200 to reflect this: "The comparable number of APEs for both dry- and wet-coupling conditions aligns with the finding of Findell and Eltahir (2003) that the SGP is located in the transitional region where negative and positive feedback days occurred with similar frequency."

- Interestingly, while working on the project that eventually led to Findell et al. (2011), I searched at length for an improvement to $HI_{low}$, trying to determine the best atmospheric levels to consider for a humidity deficit metric. In the end, I did not find a perfect level or set of levels. Instead, I found that *some* measure of the humidity deficit was necessary, but I could not conclude that inclusion of higher atmospheric levels would improve on the insights gained from $HI_{low}$. Your work clearly indicates that there are times when higher-level moisture information is needed; I wonder if other times the higher-level information actually muddies the water. (No recommended action here, just some interesting things to think about.)

  Thank you for the comment! This is a really good point. The challenges you mentioned in determining the most effective atmospheric levels for a humidity deficit metric resonate with the complexities we encountered in our research. We recognize, as you pointed out, that incorporating higher-level atmospheric moisture information into the LAC analysis, although beneficial in certain contexts, may be also obfuscating in others. While our current approach focuses on assessing the relative importance of LT humidity across different groups (dry- vs. wet-coupling), your comment highlights the nuanced nature of this analysis. It underscores the need to weigh the value of higher-level data against its potential to complicate the overall understanding. We appreciate your suggestion to consider these complexities. Although no direct action may stem from this immediately, we will explore ways to integrate this perspective into our future research directions.

- On line 148 you indicate that "a strong increase in moist static energy from the land surface moisture" is the same as a small $HI_{low}$. This is not accurate. That quoted phrase is consistent with a large latent heat flux.

  Thanks. We have revised this accordingly.

  Lines 175-177: "In this framework, wet soil advantage regime occurs when the atmospheric state is closer to the wet adiabatic rate, resulting in a low CTP and large latent heat flux (small $HI_{Low}$)."

- Around this point in the paper (~line 148), it became clear that your treatment of small $HI_{low}$ values as a surrogate for wet soil advantage days neglects the portion of the CTP-$HI_{low}$ framework that is so humid (such low $HI_{low}$ values) that any surface flux partitioning can trigger convection (Labeled "Atmospherically controlled days with convection over wet or dry soils" in the CTP-$HI_{low}$ framework schematic).

Thank you for this comment. We agree that our approach to categorizing dry-/wet-coupling APEs might have overlooked certain APEs with low $HI_{low}$ values, specifically those not associated with sufficiently high soil moisture. To address this, we have incorporated a supplementary figure (Fig. S1, attached below) that illustrates the distribution of $HI_{low}$ and FWI for these cases. Additionally, we have introduced a new category named "other APEs" in that figure. Related discussions are added at Lines 200-207:

"In addition, our analysis also shows that (Figure S1), within all APEs, 16 instances exhibit a $HI_{Low}$ lower than 5°C — a threshold established in Findell and Eltahir (2003a, 2003b). Among these, 8 are wet-coupling APEs and have significantly higher FWI compared to other groups. This suggests that the low $HI_{Low}$ values observed before noon in these cases are likely influenced by soil moisture evaporation, rather than being purely controlled by atmospheric factors. Furthermore, one of these cases is categorized as dry-coupling APE, and seven as "other APEs", which are APEs not categorized as either dry-coupling or wet-coupling APEs. These cases likely represent "atmospherically controlled days", as per the CTP-$HI_{Low}$ framework, and only account for a small fraction (~2.2%) of all APEs we identified."

[Figure]

Figure S1: (a) Distribution of CTP versus $HI_{Low}$ for three APE categories: wet-coupling (blue), dry-coupling (red), and other (black) APEs. (b) Distribution of FWI for the wet-coupling APEs, subdivided into two groups based on $HI_{Low}$ values: $HI_{Low} < 5°C$ and $HI_{Low} \geq 5°C$. (c) Same as b, but for dry-coupling APEs. (d) Same as b, but for other APEs. In each boxplot in (b)-(d), the box represents the interquartile range (IQR), which spans from the first quantile (Q1) to the third quantile (Q3) of the sample; the red line inside the box represents the median value; value larger than Q3+1.5×IQR or smaller than Q1-1.5×IQR is regarded as outlier and marked as a hollow dot; the whiskers extends to the furthest value that is not an outlier.

- Figure 4 has only 1 panel, but in the paragraph beginning on line 216 you discuss panels a, b, and c. I think you mean to refer to different line types.

  Thanks for spotting this error. Those were referring to an earlier version of that figure which has three panels. We have removed "a", "b", "c" in that paragraph.

- Final paragraph of the results section: You talk about differences between deep, shallow and convective congestus. I think you should show some figures to help highlight these results.

The result is now shown in Fig. S2 (attached below). We also revised Lines 307-308 to better reflect this:

"In addition, we evaluate the variations of deep- (cloud top height (CTH) > 8 km), shallow- (CTH <3km), and convective congestus (CTH between 3 km and 8 km) associated with APEs based on hourly precipitation and cloud fraction following Zhuang et al. (2017). In general, APEs associated with all three convective types increase with BLT under dry-coupling conditions (Figure S2). Under wet-coupling condition, APEs associated with deep convection does not exhibit a clear dependence on BLT. However, APEs associated with shallow convection decreases with increasing BLT, while those associated with congestus increase with increasing BLT. These results imply that the increase in BLT can lead to a deepening of shallow convection into congestus due to reduced buoyancy dilution caused by entraining wetter LT air for wet-coupling convection."

[Figure]

Figure S2: Distribution of (a) deep convection (DC), (b) shallow convection (SC), and (c) congestus APEs over wet- and dry- coupling conditions as a function of BLT percentile with every 0.2 bins. Their correlation coefficients with BLT percentiles are shown in the legend, where one asterisk marks significance at p<0.1 and two asterisks indicate significance at p<0.05.

**References**

Findell, Kirsten L., and E A B Eltahir, 2003b: Atmospheric controls on soil moisture-boundary layer interactions. Part II: Feedbacks within the continental United States. *Journal of Hydrometeorology*, 4(3), 570-583.

Findell, Kirsten L., Pierre Gentine, Benjamin R Lintner, and Christopher Kerr, June 2011: Probability of afternoon precipitation in eastern United States and Mexico enhanced by high evaporation. *Nature Geoscience*, 4(7), DOI:10.1038/ngeo1174.

---

## Author Comment (AC2)

**Reviewer 2 (Anonymous):**

Review of "Influence of Lower Tropospheric Moisture on Local Soil Moisture-Precipitation Feedback over the U.S. Southern Great Plains" by Wang et al.

**General comments:**

This paper examines the role of lower troposphere (LT) moisture in land-atmosphere coupling (LAC) using radiosonde data from the US Southern Great Plains (SGP) site. The analysis focuses on afternoon precipitation events (APEs) in the warm season (May– September). It is found that LT moisture has a greater impact on dry-coupling APEs than on wet-coupling APEs. A higher BLT, which is a vertically integrated LT buoyancy uncoupled with the PBL humidity, tends to increase both the frequency and intensity of dry-coupling APEs. The paper is overall well-structured and easy to follow, and the findings clarify the importance of warm-season LT moisture in dry-coupling conditions in the SGP. I have a few minor suggestions for the authors to consider.

**Specific comments:**

1. While the Introduction Section provides a detailed review of previous studies of LAC and LT moisture, little is mentioned about the SGP and why this region is selected to study the impact of LT moisture. It is also not clear why ARM SGP radiosonde data were used as the primary data in this work. A brief background information about LAC and LT moisture in the SGP and a clarification of the novelty of the approach would be helpful and informative in the Introduction Section.

We have revised the last paragraph of the introduction section to address this. **Note that all line numbers listed below are referred to the track-change version of the revised manuscript.**

Line 66-74: "The Department of Energy's Atmospheric Radiation Measurement (DOE ARM) project has been pivotal in providing comprehensive datasets for investigating land-atmospheric interactions over the past two decades (e.g., Zhang and Klein, 2010; Santanello et al., 2018). Among the various ARM sites, the Southern Great Plains (SGP) site stands out as the project's inaugural site and one of the most heavily instrumented sites. The SGP region is also widely known as a hotspot of land–atmosphere interactions, as evidenced by numerous past research (e.g., Wakefield et al., 2019; Santanello et al., 2018; Dirmeyer and Koster, 2006; Koster, 2004; Koster et al., 2006; Guo et al., 2006). This study aims to quantify the impact of LT humidity on the SM-P relationship and local LAC at the SGP site by utilizing an entrained parcel buoyancy model (Zhuang et al. 2018) and the correlation between LT humidity and near-surface humidity."

2. The concept of the lower troposphere (LT) was brought up very early in the paper, but it is not clearly defined until section 2.2 (line 135). It would be better to clarify the definition of LT in the earlier part of the paper.

Done! We have defined LT earlier in the introduction (Line 52).

Lines 51-53: "However, the humidity in the lower troposphere (LT) above the PBL, i.e., ~2-4 km above ground level (AGL), is not explicitly included in previous research."

3. It would be nice to have a section to discuss the uncertainties associated with the data and/or methodology. For instance, the coupled LT humidity profile is reconstructed by linear regressions. What's the uncertainty

of the approach? Are all the APEs convective precipitation? In addition, it also would be interesting to briefly discuss to what extent the approach used here and findings in the SGP can be generalized to other regions.

Thank you for pointing out potential limitations/uncertainties in this study. We have revised the last paragraph of the manuscript to reflect these points.

Lines 356-368: "However, there are still limitations in our work. One key concern is the potential uncertainties introduced by constructing the land-coupled LT humidity profile via linear regression. Such uncertainties arise mainly from the linear model's inherent assumptions, including the constancy of relationships under varying conditions and the potential oversight of non-linearity. A more thorough investigation into the model's residuals and additional sensitivity analyses could provide deeper insights into these uncertainties. Furthermore, our categorization of APEs may not be always associated with convective precipitation, given that it relies solely on the region-average precipitation data. Improving the classification of APEs, possibly by integrating convection classification results from radar observations, could lead to more precise interpretations. This study only focuses on a single location, i.e., SGP, thus expanding research to include a variety of climatic zones would be crucial in assessing the broader applicability of our methods and conclusions. Our future work will also involve investigating the primary source of LT humidity and employing both $B_{LT}$ and CTP/HI$_{Low}$ as atmospheric indicators to identify global regions with diverse LT humidity-SM-P relationships, thereby advancing our understanding of LAC on a broader scale."

4. In section 2.1, it would be nice to provide more details about the SGP CF site, such as location (lat, long), data coverage, and why the site was selected.

We added some more details in the introduction (Lines 66-74) and in section 2.1 (Lines 80-82).

Lines 66-74: "The Department of Energy's Atmospheric Radiation Measurement (DOE ARM) project has been pivotal in providing comprehensive datasets for investigating land-atmospheric interactions over the past two decades (e.g., Zhang and Klein, 2010; Santanello et al., 2018). Among the various ARM sites, the Southern Great Plains (SGP) site stands out as the project's inaugural site and one of the most heavily instrumented sites. The SGP region is also widely known as a hotspot of land–atmosphere interactions, as evidenced by numerous past research (e.g., Wakefield et al., 2019; Santanello et al., 2018; Dirmeyer and Koster, 2006; Koster, 2004; Koster et al., 2006; Guo et al., 2006). This study delves into the impact of LT humidity on the SM-P relationship at the SGP site, aiming to determine its influence on the local LAC."

Lines 81-83: "Unless stated otherwise, all measurements are taken at the DOE ARM SGP central facility (CF) in north-central Oklahoma (36.60°N, 97.48°W), and the region within a 50-km radius of the CF for 2001-2018."

5. In sections 2.1.1-2.1.3, please add information about the temporal resolution and coverage of the datasets and variables used in the paper. It also would be informative to discuss the error ranges of the data, if possible.

We have revised the sections accordingly to show the temporal resolution and coverage information.

Section 2.1.1 – Sounding profiles: Line 86-89: "This data is available four times daily at 05:30, 11:30, 17:30, and 23:30 local standard time (LST). We only use the 11:30 LST sounding data as it best represents the precondition of afternoon convection. "

Section 2.1.2 – Soil moisture: Lines 100-104: "we used FWI at 25 cm measurement depth provided by the Oklahoma Mesonet Soil Moisture (OKMSOIL) value-added product (VAP) (available at https://www.arm.gov/capabilities/vaps/okmsoil). This data has a 30-min resolution, and we use the average FWI during 06:00-12:00 LST to represent soil moisture condition before afternoon precipitation at daily scale."

Section 2.1.3 – Precipitation: Lines 108-112: "The Arkansas-Red Basin River Forecast Center (ABRFC) precipitation data is based on WSR-88D Nexrad radar precipitation estimates and rain gauge reports with extensive quality control (Fulton et al., 1998). This is an hourly gridded data product and is available at https://www.arm.gov/capabilities/vaps/abrfc. We used spatially averaged data over the region within a 50 km radius of the SGP CF for this study."

6. Section 2.1.4, can you please provide the equation used to calculate PBL height?

Sorry for the confusion. We did not calculate the PBL, instead, it's an ARM value added product (VAP). We revised the description to better reflect this:

Lines 114-118: "PBL height data are obtained from the ARM's Planetary Boundary Layer Height (PBLHT) value-added products derived from radiosonde data using the algorithm developed by Liu and Liang (2010). This data is available at https://www.arm.gov/capabilities/vaps/pblht."

We did not list the equations used by Liu and Liang (2010) as they involve many steps and it's not the focus of this study.

7. Line 116, how do you determine the height of the mixed layer?

In this study, the mixed layer is simply defined as 0-1km AGL. This information is added at Line 128.

8. Section 2.4, are both dry- and wet-coupling defined on the daily time scale?

Yes, dry-/wet- coupling is defined on daily scale. We have revised Lines 182-183 to clarify this.

"We first calculate CTP and $HI_{Low}$ using sounding data at 11:30 LST, and average FWI during 06:00-12:00 LST. Then dry-coupling cases are defined as days with anomalously high CTP …"

9. I suggest moving Figure S1 to the main text, as it provides useful information and there is sufficient room for one more figure in the main text.

Done. Figure S1 is now Figure 6.

10. Fig. 1, please consider marking the coefficients that are significant at the 95% confidence level. The caption mentioned "…value of 18 years", but 2001-2019 (line 74) is 19 years.

All correlation coefficients in Fig. 1 (0-4 km, all 18 years) are significant at 0.05 level. We have added this information in the caption.

And sorry for the typo, we use 2001-2018 data (18 years in total). We have corrected this throughout the manuscript.

11. Figs. 2-3, consider marking profiles where the composite differences are significant.

Thanks for the suggestions. We have modified Figs. 2-3 to mark significance (by thicker lines).

[Figure]

Figure 2: Composite difference of a) temperature ($T_{diff}$), b) specific humidity ($q_{diff}$), and c) relative humidity ($RH_{diff}$) profiles between APEs and non-APEs for dry- (red lines) and wet- (black lines) coupling cases. The thicker portions of the lines indicate where the differences are statistically significant at 0.05 level.

[Figure]

Figure 3: Composite difference of a) temperature ($T_{diff}$), b) specific humidity($q_{diff}$), and c) relative humidity ($RH_{diff}$) between dry- and wet-coupling APEs (Dry minus Wet). The thicker portions of the lines indicate where the differences are statistically significant at 0.05 level.

12. Fig. 4, why is the data coverage 2001-2018 instead of 2001-2019?

Sorry for the mistake, we actually use 2001-2018 data for this study (18 years in total). We have corrected this in the revised manuscript.

**Technical corrections:**

1. In lines 216 and 219, there is no "Figure 4a" nor "Figure 4b".

Thanks for pointing this out. Those were referring to an earlier version of that figure which has three panels. We have removed "a", "b", "c" in that paragraph.

2. In Figure 5, the last label of the x-axis ("200") is too close to the first label of the vertical axis of the bar plot ("0"), so it looks like "2000".

Thanks, we have updated Fig. 5 accordingly.

---

## Author Response (AR2)

Dear Editor and Reviewers,

We sincerely appreciate your acceptance of our manuscript. In the final version, we have included "code availability," "data availability," and "author contributions" as required by the submission system.

Best,
Yizhou Zhuang and Co-authors